# The Utility of Conventional Amino Acid PET Radiotracers in the Evaluation of Glioma Recurrence also in Comparison with MRI

**DOI:** 10.3390/diagnostics12040844

**Published:** 2022-03-29

**Authors:** Giulia Santo, Riccardo Laudicella, Flavia Linguanti, Anna Giulia Nappi, Elisabetta Abenavoli, Vittoria Vergura, Giuseppe Rubini, Roberto Sciagrà, Gaspare Arnone, Orazio Schillaci, Fabio Minutoli, Sergio Baldari, Natale Quartuccio, Sotirios Bisdas

**Affiliations:** 1Nuclear Medicine Unit, Department of Interdisciplinary Medicine, University of Bari Aldo Moro, 70124 Bari, Italy; giuliasanto92@gmail.com (G.S.); anna.giulia.nappi@gmail.com (A.G.N.); giuseppe.rubini@uniba.it (G.R.); 2Nuclear Medicine Unit, Department of Biomedical and Dental Sciences and Morpho-Functional Imaging, University of Messina, 98125 Messina, Italy; riclaudi@hotmail.it (R.L.); fminutoli@unime.it (F.M.); sbaldari@unime.it (S.B.); 3Nuclear Medicine Unit, Department of Experimental and Clinical Biomedical Sciences “Mario Serio”, University of Florence, 50134 Florence, Italy; flavialinguanti@hotmail.it (F.L.); elisabettabenavoli@gmail.com (E.A.); vittoriavergura@libero.it (V.V.); roberto.sciagra@unifi.it (R.S.); 4Nuclear Medicine Unit, A.R.N.A.S. Ospedali Civico, Di Cristina e Benfratelli, 90127 Palermo, Italy; gaspare.arnone@arnascivico.it (G.A.); natale.quartuccio84@hotmail.it (N.Q.); 5Department of Biomedicine and Prevention, University of Tor Vergata, 00133 Rome, Italy; orazio.schillaci@uniroma2.it; 6Department of Neuroradiology, The National Hospital for Neurology and Neurosurgery, University College London NHS Foundation Trust, London WC1N 3BG, UK

**Keywords:** glioma, MRI, PET, FET, MET, DOPA, recurrence

## Abstract

Aim: In this comprehensive review we present an update on the most relevant studies evaluating the utility of amino acid PET radiotracers for the evaluation of glioma recurrence as compared to magnetic resonance imaging (MRI). Methods: A literature search extended until June 2020 on the PubMed/MEDLINE literature database was conducted using the terms “high-grade glioma”, “glioblastoma”, “brain tumors”, “positron emission tomography”, “PET”, “amino acid PET”, “[^11^C]methyl-l-methionine”, “[^18^F]fluoroethyl-tyrosine”, “[^18^F]fluoro-l-dihydroxy-phenylalanine”, “MET”, “FET”, “DOPA”, “magnetic resonance imaging”, “MRI”, “advanced MRI”, “magnetic resonance spectroscopy”, “perfusion-weighted imaging”, “diffusion-weighted imaging”, “MRS”, “PWI”, “DWI”, “hybrid PET/MR”, “glioma recurrence”, “pseudoprogression”, “PSP”, “treatment-related change”, and “radiation necrosis” alone and in combination. Only original articles edited in English and about humans with at least 10 patients were included. Results: Forty-four articles were finally selected. Conventional amino acid PET tracers were demonstrated to be reliable diagnostic techniques in differentiating tumor recurrence thanks to their high uptake from tumor tissue and low background in normal grey matter, giving additional and early information to standard modalities. Among them, MET–PET seems to present the highest diagnostic value but its use is limited to on-site cyclotron facilities. [^18^F]labelled amino acids, such as FDOPA and FET, were developed to provide a more suitable PET tracer for routine clinical applications, and demonstrated similar diagnostic performance. When compared to the gold standard MRI, amino acid PET provides complementary and comparable information to standard modalities and seems to represent an essential tool in the differentiation between tumor recurrence and other entities such as pseudoprogression, radiation necrosis, and pseudoresponse. Conclusions: Despite the introduction of new advanced imaging techniques, the diagnosis of glioma recurrence remains challenging. In this scenario, the growing knowledge about imaging techniques and analysis, such as the combined PET/MRI and the application of artificial intelligence (AI) and machine learning (ML), could represent promising tools to face this difficult and debated clinical issue.

## 1. Introduction

Glioma are the most common intra-axial primary tumors of the central nervous system (CNS) arising from glial cells, with an estimated annual incidence approximately of six cases per 100,000 individuals worldwide [1]. For the past century, the classification of brain tumors has been largely based on their microscopic similarities including different cells’ origins and their presumed levels of differentiation. The 2007 World Health Organization (WHO) introduced a new classification system according to the characteristics of the anaplasia or the presence of nuclear atypia, mitosis, endothelial proliferation, and necrosis. Glioma were divided into four subgroups: “low-grade glioma” (LGG) to indicate WHO grade I and II glial tumors, and “high-grade glioma” (HGG) or “malignant” for grade III and IV tumors [2]. The revised version of the 2016 WHO classification of glioma integrated histologic and molecular findings to provide a much more accurate prognostic value than the previous one. Notably, the presence/absence of the IDH 1–2 gene mutation, as the presence/absence of the codeletion of chromosomes 1q-19q are now considered as determining factors in the definition of the different histo-molecular subtypes [3]. Despite treatments, these tumors exhibit a poor prognosis with a median overall survival of 15 months for glioblastoma multiforme (GBM), the most frequent glioma type in adults (46%) and the most lethal. This poor prognosis partly results from a lack of significant advances in treatment prolonging life and a high rate of recurrence/progression with a median progression-free survival (PFS) of only 8 to 11 weeks for recurrent HGG [4].

### 1.1. Posttreatment Evaluation

Glioma’s treatment choice is based on different factors such as the histologic grade, location, tumor resectability, and patient’s performance status. The standard treatment for adult gliomas usually involves maximal safe resection, defined as resection of the enhancing tumor as much as possible to improve survival [5]. For diffuse glioma, resection with clear margins is virtually impossible because of their highly infiltrative nature determining the persistence of neoplastic cells in the macroscopically normal-appearing brain tissue. After surgery or in unresectable cases, patients undergo adjuvant RT and CHT with temozolomide or nitrosourea drugs (procarbazine, lomustine, vincristine). For some selected patients, immune or target therapies are also considered [6,7]. Antiangiogenic agents, such as bevacizumab, a monoclonal antibody directed against vascular endothelial growth factor (VEGF), are considered second-line treatments [8], usually reserved for recurrent disease [9]. In 2010, the treatment evaluation criteria of gliomas was revised by the Response Assessment in Neuro-Oncology (RANO) Working Group, taking into account advanced imaging approaches such as perfusion magnetic resonance imaging (PWI), magnetic resonance spectroscopy (MRS), or positron emission tomography (PET), and new standardized response criteria were developed [10]. In this scenario, some entities such as pseudoprogression (PSP), radiation necrosis (RN), and pseudoresponse were introduced to better distinguish recurrence from treatment-related changes (TRC) on imaging. PSP can be identified as an increase in contrast-enhancing and perilesional oedema in the absence of true disease progression, as a probable consequence of transiently increased permeability of the tumor and inflammation induced by radiation therapy and further increased by temozolomide [11]. Within the first 12 weeks after radiotherapy when PSP is most prevalent, real disease progression could be determined if the majority of the new enhancement is outside of the radiation field or if there is pathologic confirmation of progressive disease, according to RANO. Moreover, PSP can be confirmed if the sum of the products of perpendicular diameters between the first postradiotherapy scan and the scan at 12 weeks (or later) has not increased over 25% [10]. Usually, PSP occurs in the first 3 months after concurrent chemoradiation therapy (early PSP) [12], but it has also been described later than 12 weeks, after the end of therapy (late PSP), making the diagnosis more difficult [13]. PSP resolution occurs within a few weeks or months, and subsequently, no specific treatment is needed; therefore, these patients are at risk of inappropriate further therapy. RN is a later and chronic complication, secondary to any technique of radiation therapy, that may occur 3–12 months after the end of therapy, but also years and even decades afterward, particularly after high-dose radiotherapy. Conversely to PSP, RN does not always subside [14]. Pseudoresponse is instead associated with antiangiogenic treatments, which could produce a decrease in contrast enhancement, resulting in an apparent radiological response. For this reason, RANO criteria suggest that radiological responses should persist for at least 4 weeks to be considered as real responses [10,15]. Since therapeutic strategies and patient management for these pathological entities are fundamentally distinct, differentiation between recurrent glioma and TRC is crucial and can be challenging as both share clinical symptoms and imaging characteristics.

### 1.2. Magnetic Resonance Imaging (MRI)

Conventional magnetic resonance imaging (MRI), with the addition of contrast enhancement (ceMRI), is the method of choice for diagnosis, treatment planning, and posttreatment follow-up of brain tumors. According to RANO criteria [10], progressive disease 12 weeks after chemoradiotherapy completion on MRI imaging is characterized by:clinical deterioration (not attributable to other non-tumor causes and not due to steroid decrease);25% or more increase in the sum of the products of perpendicular diameters between the first postradiotherapy scan and the scan at 12 weeks or later;increase (significant) in non-enhancing FLAIR/T2W lesions, not attributable to other non-tumor causes;any new contrast-enhancing lesion outside of the radiation field.

Despite the introduction of RANO criteria, diagnosis of glioma recurrence remains challenging mainly due to TRC that could impact MRI findings, regardless of the time of evaluation. Notably, ceMRI non-specifically reflects the vascular surface area, and the permeability of the contrast agent across the disrupted blood–tumor barrier (BBB) can be influenced by treatments such as corticosteroid, antiangiogenic, or immunotherapy agents as well as “radiation” effects such as demyelination, ischemic injury, and oedema [16]. Therefore, the contrast enhancement on ceMRI can unspecifically increase in RN and PSP because the BBB damage mimics glioma recurrence or tumor progression. Furthermore, oedema and necrosis induced by radio- and chemotherapy and postoperative reaction could also be misinterpreted as a disease progression because of the increase in T2/FLAIR signal on MRI [17]. Recently, advanced MRI techniques such as MRS, diffusion-weighted imaging (DWI), as well as PWI have been introduced to improve diagnostic performance for the differentiation of TRC from progression [18,19,20]. However, the differentiation of these two entities is not unequivocal. Hence, there is a need for a reliable imaging technique that can be more useful to differentiate treatment-induced changes, avoiding unnecessary treatment or, on the other hand, delayed treatment of recurrence.

### 1.3. Amino Acid Tracer Positron Emission Tomography

Advanced imaging techniques such as positron emission tomography (PET) can provide a quantitative assessment of functional and metabolic changes of the tumor tissue, anticipating the morphological variations. In 1983, PET with amino acidic tracers was introduced in neuro-oncological practice. Over the last few decades, the increasing knowledge about functional imaging using amino acids PET pointed out their usefulness in overcoming the drawbacks for the detection of PSP and RN [12].

The rationale for the reliability of amino acid PET belongs firstly to their cellular uptake. Namely, cellular accumulation of these tracers is mainly driven by the activity of System L amino acid transporters (LAT1 and LAT2) that carry these amino acids into the tissue with unique metabolic pathways that can be exploited in tumor imaging [21,22]. The density of LAT expression on the cell membrane surface has been shown to be related to amino acid PET tracer uptake [23]. This metabolic mechanism is highly specific for tumor cells and very rarely BBB breakdown-influenced tracer uptake [24], resulting largely as independent from BBB treatment-induced alteration and, subsequently, in excellent tumor-to-background contrast.

After entering tumor cells, standard amino acids are mostly used for protein synthesis. Since these tracers, such as standard methionine and tryptophan analogues, seem to produce a number of non-protein-bound metabolites that make difficult protein synthesis rates, most amino acid radiotracers used for cancer imaging have been modified by adding methyl or ethyl groups to create derivatives that are less likely to be substrates for protein synthesis or other metabolic pathways [25,26]. In the past, the most widely used tracer for amino acid PET was methyl-l-methionine (MET), an essential amino acid labelled with a carbon-11 positron-emitting isotope, but the short half-life of 11C (20 min) limits the use of MET PET to centers with on-site cyclotron. For this reason, [^18^F]labeled amino acids, such as [^18^F]fluoro-l-dihydroxy-phenylalanine (FDOPA) and [^18^F]fluoroethyl-tyrosine (FET), were developed to provide a more suitable PET tracer for routine clinical applications [27,28,29] and for research purposes [30]. In 2018, the revised version of the practice guidelines for the imaging of gliomas using PET with radiolabeled amino acids and [^18^F]Flurodeoxyglucose-[^18^F]FDG PET was developed from the cooperative work of the European Association of Nuclear Medicine (EANM), the Society of Nuclear Medicine and Molecular Imaging (SNMMI), the European Association of Neuro-Oncology (EANO), and the working group for Response Assessment in Neuro-Oncology with PET (PET-RANO). These standards/guidelines aim to guide PET findings’ interpretation to improve the feasibility of a PET tracer in neuro-oncological practice, providing detailed acquisition protocols, and visual and semiquantitative analysis as well as a PET parameters’ threshold, although the literature is not uniform and evidence is still in progress [16]. This comprehensive review represents an update of the most recent evidence about conventional amino acid PET radiotracers in the challenging scenario of differential diagnosis between glioma progression/recurrence and TRC. Moreover, we aimed to compare amino acid PET with the gold standard MRI to provide an integrated method, using the best-advanced imaging modality in the detection of recurrence in glioma patients.

## 2. Search Strategy

A PubMed/MEDLINE search of the published literature with a combination of the search terms “high-grade glioma”, “glioblastoma”, “brain tumors”, “positron emission tomography”, “PET”, “amino acid PET”, “[11C] methyl-l-methionine”, “[^18^F]fluoroethyl-tyrosine”, “[^18^F]fluoro-l-dihydroxy-phenylalanine”, “MET”, “FET”, “DOPA”, “magnetic resonance imaging”, “MRI”, “advanced MRI”, “magnetic resonance spectroscopy”, “perfusion-weighted imaging”, “diffusion-weighted imaging”, “MRS”, “PWI”, “DWI”, “hybrid PET/MR”, “glioma recurrence”, “pseudoprogression”, “PSP”, “treatment-related change”, and “radiation necrosis” from 2000 until June 2021 was performed. The literature search revealed 157 articles. Only original articles edited in English and about humans with at least 10 patients were included. Case reports, editorials, preclinical papers were not included. Additional literature was retrieved from the reference lists of all identified articles. After screening titles and abstracts, and reading full-texts, 44 articles were finally selected for review discussion. The study workflow is illustrated in Figure 1.

## 3. FET

[^18^F]fluoroethyl-tyrosine (FET) is an artificial amino acid taken up into upregulated tumoral cells by Na+ independent transport via the LAT system, independently of BBB leakage [31]. FET is not incorporated into proteins, as with the natural amino acids, and its uptake grade is not directly proportional to tumor differentiation status [27,32,33]. FET is diffuse in Europe and has been shown to provide high sensitivity and specificity for glioma detection and low uptake in the inflammatory and healthy brain [31,34], resulting in a reliable diagnostic tool for differentiating tumor recurrence/progression from TRC. Dynamic FET PET and time–activity curves (TACs) offer additional information on tracer kinetics. As known, in HGG, FET uptake is characterized by an early peak 10–15 min after injection, followed by a decrease in radiopharmaceutical’s uptake, similarly to recurrence [35,36]; differently, an LGG shows a typical delayed and steadily increasing tracer uptake similar to TRC [37]. These patterns are usually observed for FET PET and not for other amino acid tracers such as MET and FDOPA. Static PET scan protocols might not reveal the active metabolic tumor and might suffer from a lack of standardized acquisition protocols; thanks to the proprieties of not being metabolized after the entry into the cell, advanced pharmacokinetic analysis of TACs from dynamic FET PET scans, using compartment models, was exploited [34].

Several studies investigated the role of FET in the evaluation of recurrence and a variable diagnostic accuracy ranging between 81% and 99% was reported [38,39]. This wide range could be explained considering the different PET parameters analyzed, e.g., tumor-to-background ratios (TBR_max_ and TBR_mean_), acquisition time protocols (static vs. dynamic), the uptake kinetics (time to peak—TTP, in minutes from the beginning of the dynamic acquisition up to the maximum SUV of the lesion) as well as different patient populations, tumor subtypes, and treatments [40]. Notably, guidelines reported a common threshold to assess glioma recurrence on FET imaging defining a TBR_mean_ of 2.0 and a TTP < 45 min [16]. Using these cutoff values, Galldiks and colleagues yielded the best result for identifying tumor recurrence or progression, with a sensitivity of 93%, a specificity of 100%, and an accuracy of 93% [35]. Nevertheless, PET parameters and the relative threshold are not standardized in all studies and different diagnostic performances were reached using different values. For example, in a cohort of 26 patients with GBM, a cutoff value of 1.9 for TBR_max_ allowed differentiating between true progression and late PSP. Moreover, in the same study, the dynamic acquisition also identified different curve patterns for differential diagnosis. A FET uptake peaking at a midway point (>20–40 min) followed by a plateau or a small descent was described as pattern II; conversely, the uptake peaking early, at 20 min, followed by a constant descent was named pattern III. Curve patterns type II and III resulted in being predictive for true progression and TRC, respectively [40]. However, it is important to underline that, in clinical routine, dynamic FET PET imaging has several limitations. As known, it requires longer acquisition times (50–60 min vs. 10–20 min for a static scan), which reduces patient compliance and could cause motion artifacts, with increasing costs of the investigation [35]. Recently, Bashir et al. performed a static FET PET study in a homogenous population of 146 suspected recurrent GBM patients 20 min after administration, demonstrating that FET parameters were significantly higher in patients with recurrence compared with patients with TRC (TBR_max_, 3.2 vs. 1.6; TBR_mean_, 2.0 vs. 1.6; biological tumor volume—BTV,14.8 cm^3^ vs. 0.01 cm^3^; *p* < 0.0001). Using a threshold of 2.0 for TBR_max_, PET-based classifications of recurrent GBM or TRC were confirmed in 98.8% of patients [39]. In contrast to the reported higher performance of early FET imaging (10–20 min) for the primary diagnosis of glioma, in this setting of recurrent disease, static PET imaging acquired from 30 to 40 min demonstrated a sensitivity and specificity of 80.0% and 84.6%, respectively, at an optimal cutoff of TBR 2.07, providing slightly better discrimination than early images [36]. Disease progression assessment could also be challenging in recurrent GBM treated with bevacizumab, which is an antiangiogenic agent. Two clinical studies of patients with recurrent HGG undergoing bevacizumab therapy showed an advantage of FET PET over MRI in the early detection of tumor progression. Response to therapy as detected with PET (defined as >45% decrease in tumor volume) was also associated with longer overall and progression-free survival (OS and PFS) [41,42,43]. Although the potential role of FET PET in differentiating TRC from recurrence is well established, the methodology has yet to be standardized to define imaging protocols, as well as both the tumor and the normal brain reference regions, since the differentiation of a viable tumor from TRC is predominantly established by TBR [16]. In a 2019 study, the diagnostic performance of several analytic approaches in the setting of PSP in GBM was evaluated. All TBRs’ measures were significantly higher in patients with true tumor progression as compared with late PSP, regardless of the semiquantitative approach applied. Although these results are encouraging and the significance is promising, the need for a consistent method of background activity assessment is requested. A crescent-shaped background volume of interest (VOI), as a reproducible approach for methodological standardization, and an isocontour approach (including multiple voxels with the highest radiopharmaceutical uptake) were proposed to reduce noise, increase reproducibility, and avoid potential pitfalls of reference region definition (e.g., the inclusion of structural changes due to atrophy, trauma, or ischemia) [44,45].

### Comparison of FET PET with MRI

When FET PET is compared to MRI, the results are not uniform, but the added value of combined data was greatly demonstrated. PWI was often performed to improve the diagnostic accuracy, and the role of dynamic susceptibility contrast (DSC) PWI was demonstrated in HGG [46]. Furthermore, since the neoplastic hypervascularization in glioma might result in a relative increase in the cerebral blood volume (rCBV) compared to normal-appearing brain tissue, several studies analyzed FET PET parameters with PWI-derived parameters. Namely, a recent study addressed the diagnostic value of sequential DSC PWI and dynamic FET PET to differentiate tumor progression from TRC. The results showed rather low sensitivity of the rCBV_max_ (0.53), compared to the substantially higher FET PET sensitivity of combined static and dynamic (0.96) values. However, the high cutoff of rCBV_max_ achieved a high specificity, suggesting the additional diagnostic value of a sequential combination of both examinations [47]. Similar results were described by Göttler et al., indicating that the maximum FET uptake might depend more on high blood volumes than on the washout slope [48]. Other study groups reported the increased value of functional imaging over PWI MRI. Verger et al. observed that FET TBR_max_ was the only parameter that showed a significant diagnostic power to discriminate between TRC and progressive/recurrent glioma, while none of the PWI parameters reached significance. Even though, based on visual analysis, FET PET showed an increased uptake in 76% of recurrent glioma, PWI MRI showed signal abnormalities in only 52%. Surprisingly, in the subgroup of IDH-mutant tumors, PWI appeared to be more reliable than FET PET [49]. This data could be supported by recent evidence describing a significantly higher diagnostics accuracy of FET PET in IDH-wildtype glioma than in IDH-mutant ones. However, further studies are needed to validate these findings [38]. Another matter of discussion remains the poor spatial agreement between the two techniques, with a described considerable distance of hot spots between FET uptake and PWI within the area of tumor recurrence [50]. Moreover, the application of quantitative DWI-derived parameters is inconsistent in this scenario. Some studies reported that TRC show higher apparent diffusion coefficient (ADC) values than recurrent glioma, but some evidence demonstrated opposite results. In an analysis on a hybrid PET/MRI scanner conducted by Lohmeier et al., glioma relapse presented higher ADC_mean_ and TBR_max_ values than TRC, and both ADC_mean_ and TBR_max_ achieved reliable diagnostic performance in differentiating glioma recurrence from TRC as also reported by Pika et al. [36,51]. FET PET, PWI, and DWI data were combined by Sogani et al.: the authors reported significant moderate correlations between TBR_max_ and rCBV_mean_, and TBR_mean_ and rCBV_mean_, suggesting the presence of coupled vascularity and tumor amino acid uptake with mitotic activity and endothelial proliferation. At the same time, negative correlations between TBR_max_ and ADC_mean_, and TBR_mean_ and ADC_mean_ were described, suggesting increased FET uptake in areas of high mitotic potential and, consequently, increased cellular density, yielding lower ADC values [52]. Furthermore, PET parameters in combination with MRS data reached a high accuracy: when both the TBR_max_ was greater than 2.11 (or TBR_mean_ greater than 1.4) and the Cho/Cr ratio was greater than 1.4, an accuracy of 96.9% in diagnosing recurrent glioma was reported [53]. In Table 1, we describe the main characteristics of the studies regarding FET PET applications in glioma recurrence/differential diagnosis.

## 4. FDOPA

6-fluoro-(^18^F)-l-3,4-dihydroxyphenylalanine (FDOPA) is a neutral amino acid, transported into presynaptic neurons, where it is first converted into fluorodopamine by the aromatic amino acid decarboxylase (AAAD) enzyme, and subsequently accumulated in catecholamine vesicles. Similar to FET, FDOPA crosses the BBB through amino acid transporters (LAT1–2), independently of the BBB breakdown [54]. The [^18^F]FDOPA is a radiotracer (diffuse in the USA) exhibiting high uptake in malignant brain tumors and only minimal uptake in the normal cerebral cortex and white matter, except for physiological uptake in the striatum [55,56]. Conversely from FET, which is more selectively transported through LAT2 than LAT1, FDOPA uptake occurs through both transport systems. This could be associated with a higher risk for false-positive findings since recent studies revealed a crucial role of LAT1 in activated T cells and reported overexpression of the LAT1 in inflammation, while LAT2 is more tumor-selective [57]. The diagnostic performance of FDOPA PET imaging for predicting glioma recurrence/progression was evaluated by Hermann et al. in a large population of 110 patients with an initial diagnosis of grade III (*n* 33; 30.0%) or grade IV (*n* 77; 70.0%) disease. The authors reported a significant diagnostic accuracy of 82% (sensitivity, 89.6%; specificity, 72.4%) for FDOPA PET in distinguishing recurrent disease from TRC, resulting in an additional PFS predictive role [56]. Recently, in a 2020 study conducted by Zaragoni et al. in a population of 51 patients classified as 8 IDH-mutant astrocytomas (16%), 6 (12%) as IDH-wildtype astrocytomas, 12 (24%) as IDH-mutant, and 1p/19q co-deleted oligodendrogliomas, 22 (43%) as IDH-wildtype GBM, and 3 (6%) as IDH-mutant GBM, according to 2016 WHO classification, FDOPA exam reached a global accuracy of 96% for predicting glioma recurrence/progression at 6 months after PET. The semiquantitative evaluation of FDOPA PET was mainly based on static PET parameters, and guidelines suggest a maximum and mean tumor-to-striatum ratio (TSR_max_ and TSR_mean_) threshold of 2.1 and 1.8, respectively. Nevertheless, even for FDOPA PET, parameters and relative cutoff differ between studies. In the aforementioned study by Zaragoni et al., all static PET parameters (TBR_max_; TBR_mean_; TSR_max_; TSR_mean_; MTV) were significant univariate predictors of glioma recurrence/progression even using different cutoff values compared to reference values [58]. Nonetheless, more studies are needed to better standardize procedures.

In comparison with FDG PET, FDOPA PET resulted in being superior in the assessment of recurrent glioma and in the differentiation between tumor recurrence from RN with higher diagnostic accuracy (96.4% vs. 60.7%). Notably, in HGG, the sensitivity of FDOPA PET was higher than FDG (100% vs. 76.9%), with similar specificity (100%). Conversely, in LGG, FDOPA PET/CT showed a significantly higher diagnostic accuracy compared to FDG (90% vs. 20%) [59].

### Comparison of FDOPA PET with MRI

The first clinical study that systematically compared the diagnostic accuracy of FDOPA PET-computed tomography (CT) and ceMRI revealed the high sensitivity (100%) and specificity (88.89%) of FDOPA in recurrence detection, whereas ceMRI showed high sensitivity (92.3%) but poor specificity (44.44%). In the same study, FDOPA showed higher accuracy (97.1%) than ceMRI (80%) for both HGG and LGG. However, no significant positive correlation was shown among semiquantitative PET parameters with the grade of glioma, suggesting that the tumor grade did not significantly affect tracer uptake [60]. In another paper, FDOPA PET demonstrated higher sensitivity (82%) than ceMRI (52%) for the detection of recurrent glioma through a TBR > 2.0 and an SUV_max_ > 1.36 [61]. As for the other amino acid PET tracers, the second major issue was the poor spatial congruence between FDOPA uptake and MRI findings. A study reported a mean radial distance of 2.7 cm between FDOPA uptake and rCBV PWI hot spots, similar to FET [62]. Moreover, FDOPA PET seems to precede MRI in the local tumor recurrence detection. Namely, MRI is characterized by a small amount of ce adjacent to a neurosurgical resection, without distinguishing between postsurgical change or residual tumor, while the authors observed extensive FDOPA activity corresponding to not only the tumor region of ce, but also to the surrounding non-contrast-enhancing tumor irrespective of tumor grade [63,64]. Spatial correlation between MRI and FDOPA PET was studied by Karavaeva et al. in 29 patients with recurrent HGG. Namely, they compared the ADC on diffusion MRI with FDOPA uptake areas demonstrating that areas of elevated FDOPA uptake within ce tumor regions appeared to have a low ADC on diffusion MRI. This result is consistent with the hypothesis that regions of low ADC may reflect an active tumor. Moreover, the authors observed a significant positive correlation between the average mitotic activity within a resected enhancing tumor, as estimated from average Ki-67-positive cells and median FDOPA uptake within areas of contrast enhancement. This result supported the hypothesis that FDOPA PET uptake reflects the general mitotic activity of the tumor, as demonstrated in newly diagnosed gliomas [65]. In Table 2, we describe the main characteristics of the studies regarding FDOPA PET applications in glioma recurrence/differential diagnosis.

## 5. MET

l-[Methyl-11C]-Methionine (MET) is a radiotracer that easily crosses the intact BBB through sodium-independent L-type amino acid transporters. Cellular tumoral proliferation is associated with increased protein synthesis compared to the normal brain [66], so MET uptake is related to the degree of cell proliferation (Ki67 expression), the neovascularization and microvessel density, and the increased amino acid carrier-mediated and passive transports [67]. Thanks to its high tumor and low cortical background uptake, MET allows identification of the tumor mass, and in particular, the most biologically aggressive tumors, and their borders. However, low specificity with a high number of false positives was reported considering the increased MET uptake, also in non-neoplastic lesions, such as inflammation, infarction, hemorrhage, leukoencephalitis, and demyelination [68]. Over the last twenty years, the diagnostic performance of MET PET for identifying glioma recurrence was evaluated by several authors. In 2004, Tsuyuguchi et al., in a small sample of 11 HGG patients who underwent stereotactic radiosurgery, reported a sensitivity, specificity, and accuracy of MET PET in detecting tumor recurrence of 100%, 60%, and 82%, respectively [69]. Ten years later, D’Souza and colleagues, in a larger sample of 29 pretreated HGGs, reached a sensitivity, specificity, and accuracy of MET PET of 94.7%, 80%, and 89.6% [70]. Similar to other amino acid tracers, semiquantitative analysis was performed to determine the appropriate MET PET parameters that could help in the differential diagnosis and that could be used to reach the best diagnostic performance [71]. Notably, different cutoff values were reported for TBR simply because different image acquisition techniques, machines, processing techniques, and analysis tools were used, resulting in a wide range of sensitivity and specificity between studies. Following current guidelines, a threshold of 1.6 for TBR_max_ was used for discriminating recurrent gliomas [16]. In a retrospective study by Kits et al., a consecutive series of patients with neuropathologically confirmed recurrent brain tumors or radiation-induced changes were studied. SUV_max_ and SUV_mean_ were obtained in the lesion, in the contralateral mirror region, and in the contralateral frontal cortex; then, TBR ratios (TBR mirror and TBR cortex) were calculated. The diagnostic accuracy of the TBR_max_ mirror and TBR_max_ cortex were both high in discriminating recurrent tumor from radiation injury. Notably, TBR_max_ cortex ≥ 1.58 reached a sensitivity and specificity of 90% and 78%, respectively, while sensitivity and specificity for tumor recurrence using a TBR_max_ mirror ≥ 1.99 were 76% and 100%, respectively [72]. TBR has proved to be a discriminating parameter for tumor recurrence both using the maximum (TBR_max_) as well as the mean (TBR_mean_) parameter value [73]. However, other studies showed the superiority of TBR_mean_ over TBR_max_: Terakawa et al. performed MET PET in a large sample of 77 patients including both metastatic brain tumor and glioma. In the subgroup of 26 patients with suspected glioma recurrence, TBR_mean_ provided the best sensitivity and specificity in differentiating glioma recurrence from RN [74]. Some years later, a similar conclusion was made by Shihido et al. In their study evaluating a small but homogenous cohort of 21 grade III and IV glioma patients, the authors demonstrated that SUV_max_ did not show a significant difference between necrosis and recurrence, while TBR resulted in being significantly higher for recurrent glioma than for necrotic lesions (*p* < 0.01). These results belong to the evidence that SUV produced a high standard deviation, even in the normal grey matter, while TBR could reduce individual differences [69,75]. Interestingly, in a 2016 study conducted on 42 gliomas previously treated, the best diagnostic accuracy was reached using both TBR (*p* = 0.009) and MTV (*p* = 0.001) with the optimal cutoff values of 1.43 and 6.72 cm^3^, respectively [66]. When compared with other radiopharmaceuticals, MET was shown to be more reliable than FDG in detecting tumor recurrence, irrespective of grade, achieving a sensitivity of 94.7% and a specificity of 88.8%, with less interobserver variability and better delineation of tumor extension (vs. 81.2% and 88.9% for sensitivity and specificity, respectively, using FDG) [76]. MET PET was also compared with FET in a 2011 study: Grosu et al. did not find any significant difference between MET and FET in the diagnosis of recurrent glioma, resulting in high sensitivity (91%) and specificity (100%) of both tracers for the differentiation of tumor from TRC/PSP [28].

### Comparison of MET PET with MRI

Similar diagnostic performances for both MET PET and advanced MRI were reported in the literature, yielding the highest accuracy when they were combined. In a recent study, SUV_max_, SUV_mean_, TBR, and rCBV_mean_ resulted in being significantly higher for patients with recurrence than for patients with radiation injury, with concordance on both MET PET/CT and PWI MRI in differential diagnosis [77,78]. D’Souza et al. reached similar conclusions comparing MET PET/CT and advanced MRI (MRS and PWI) in 29 patients with HGG: the authors indicated that MET PET seemed more sensitive (94.7% vs. 84.2%) and advanced MRI imaging more specific (90% vs. 80%), but no statistically significant difference in the diagnostic performance of either technique was observed [70]. Based on the RANO classification, sensitivity, specificity, and positive predictive value were calculated for MRI alone resulting in 86.1%, 71.4%, and 88.6%, respectively. The same values were calculated for [^11^C]MET PET reaching 96.7%, 73.7%, and 85.7%, respectively. When both imaging modalities were integrated, [^11^C]MET PET/MRI reached the highest sensitivity (97.1%), specificity (93.3%), and PPV (97.1%). Diagnostic accuracy was 82% for MRI, 88% for MET PET, and 96% for hybrid MET PET/MRI. A significant difference was found among hybrid MET PET/MRI and MRI (*p* = 0.008), whereas no significant difference was found among hybrid MET PET and MRI alone (*p* = 0.021) or MET PET/MRI and MET PET alone (*p* = 1) [72]. In Table 3, we describe the main characteristics of the above-mentioned studies regarding MET PET applications in glioma recurrence/differential diagnosis.

## 6. Other Amino Acid Transporters for Future Directions

In the challenging scenario of amino acid PET tracers, a debate field of interest is represented by the non-natural amino acid compounds. This attention is growing from the evidence that the “in vivo” tumor uptake of radiolabeled amino acids depends mainly on the rates of amino acid transport rather than protein synthesis. In mammalian cells, more than 20 distinct amino acid transporters have been identified, which differ for substrate specificity and sodium, and other ions’ dependency, pH sensitivity, and transport mechanism [79]. The most recognized transport systems include system L, system A, and system ASC. All of them are sodium-dependent transporters, except for system L targeted by the majority of radiolabeled amino acids for tumor imaging, as described above [80,81,82].

An important limitation of system L transport substrates is the inability to directly concentrate substrates intracellularly. This factor could be associated with relatively low tumor-to-tissue ratios, which can reduce the sensitivity of the amino acid tracers. Conversely, other amino acid transporters, such as system A, can concentrate substrates intracellularly, providing higher and persistent tumor uptake. However, the lack of activity of many amino acid transporters at the luminal side of the BBB allows access of their substrates only to the enhancing regions of brain tumors [83].

These observations prompted Bouhlel and colleagues in 2015 to develop ^18^F-labeled amino acid tracers that target both system L and non-system L amino acid transporters to exploit both the system L transport and the intracellular concentration provided by non-system L transporters. Based on previous data, the authors proposed a new ^18^F-labeled analogue, 2-amino-5-[^18^F]fluoro-2-methylpentanoic acid ([^18^F]FAMPe), with longer alkyl chain lengths to increase recognition by system L transporters [84,85,86,87]. However, they showed that the longer side chain failed to be recognised by system A transporters, but the (S)-FAMPe enantiomer appeared to be mediated in part by the glutamine transporter (ASC). In addition, compared with other compounds, the authors showed that both enantiomers of [^18^F]FAMPe had a higher tumor-to-brain ratio compared to (S)-[^18^F]FET (*p* < 0.001) but lower than (R)-[^18^F]MeFAMP. These findings suggested that [^18^F]FAMPe could provide better tumor visualization, particularly useful in monitoring therapy response. However, the high tumor-to-brain ratios due to low brain uptake could decrease the visualization of non-enhancing gliomas regions, representing a limit of the tracer [87].

Another ^18^F-labeled non-natural amino acid is represented by (S)-2-amino-3-[1-(2-^18^F-fluoroethyl)-1H-[1,2,3]triazol-4-yl]propanoic acid ([^18^F]AFETP) [80]. This tracer is a structural analog of histidine and showed promising preclinical results in the rat 9 L gliosarcoma model. Interestingly, the in vitro uptake of this compound was mediated, in part, by cationic AA transport [88,89]. The study by Sai et al. using [^18^F]AFETP in mice, demonstrated higher uptake in tumor than in most normal tissues. Comparing [^18^F]FDG, [^18^F]FET, and [^18^F]AFETP, the latter provided the best brain tumor visualization mainly due to a lower normal brain uptake [80,89].

As already described in the “Amino Acid Tracer Positron Emission Tomography” section, higher tumor-to-normal tissue ratios can provide better tumor visualization and a larger dynamic range for assessing response to therapy. However, most low-grade tumors do not have grossly disrupted BBBs, and many high-grade gliomas, including glioblastoma, have non-enhancing regions not readily assessed with conventional contrast-enhanced MRI. For this purpose, system L substrates such as [^18^F]FET, thanks to their ability to cross the BBB, could image the non-enhancing regions of gliomas more than conventional MRI or [^18^F]FDG [90,91]. Similarly, [^18^F]AFETP using the cationic AA transporter CAT-1, active at the BBB, could allow visualization of the non-enhancing regions, opening new opportunities more than [^18^F]FET in a very heterogeneous disease such as glioma [80].

## 7. Innovative Approaches

Considering the recent advances in medical image analysis, artificial intelligence (AI) and machine learning (ML) have also gained increasing attention in the field of neuro-oncology [92]. In the challenging scenario of recurrent gliomas, only a few studies that applied these innovative techniques using amino acid PET are available. However, their results deserve particular attention. Despite the promising value of conventional PET parameters to reflect metabolism, PET tracer uptake depends on several physiological features, such as perfusion, cell proliferation, tumor viability, hypoxia, and aggressiveness, that could reflect tumor uptake heterogeneity [93]. In a 2016 pilot study, Kebir et al. analyzed 14 histologically proven HHG treated with chemoradiotherapy before FET PET. A set of 19 conventional and textural FET PET features were evaluated and subjected to unsupervised consensus clustering. Using the nearest shrunken centroid method called PAM, FET PET features were identified and associated with three different clusters: cluster 2 was associated with high values of the textural characteristics (Contrast and Entropy) and designated as “high heterogeneity cluster”. Cluster 3 was largely associated with inverse loadings of FET PET textural features as compared with cluster 2, and named “low heterogeneity cluster”. Cluster 1 had the least variability in features compared to clusters 2 and 3 and was defined as an “intermediate cluster”. Cluster 3 provided high sensitivity and specificity (90% and 75%, respectively) for detecting true progression with a negative predictive value (NPV) of 75% [94]. A comparison with conventional FET PET receiving operator characteristic (ROC) curve analysis and ML algorithms was conducted by the same group. A cohort of 44 IDH-wildtype GBM patients was examined using an ML model based on the linear discriminant analysis (LDA) approach. The authors compared the AUC for LDA ML with that of TBR_max_ and TBR_mean_ and a combination of TBR_max_, TBR_mean_, and TTP. A significant difference was shown between the AUC of LDA ML compared to TBR_mean_ (*p* = 0.035), but none compared to those of TBR_max_ (*p* = 0.081) and the combination of TBR_max_, TBR_mean_, and TTP (*p* = 0.132) [95]. Lohmann et al. also investigated the potential of textural features of FET PET for discriminating between PSP and tumor progression. Thirty-four glioblastoma patients with MRI findings suspicious for tumor progression within the first 12 weeks after completion of chemoradiation were included. Conventional static and dynamic PET parameters and four selected radiomics features were evaluated. The final ML model showed 70% accuracy in the test dataset and correctly identified all patients with PSP [96]. To emphasize the emerging potentiality of multiparametric FET PET/MRI even in the era of AI, Paprottka et al. described a fully automated pipeline, from longitudinal tumor segmentation and features extraction to classification. The study integrated information from FET PET, DSC-derived CBV maps of PWI MRI, and amide proton transfer-weighted (APTw) imaging, a relatively novel molecular MRI technique. Sixty-six patients were finally included. For modeling data, the authors used a random forest approach, which is a well established ML model for classification in the presence of (potentially) correlated input data. ROC analysis for the identification of disease progression in the fully automated data analysis yielded an AUC of 0.85, with an accuracy of 0.86 (sensitivity 0.91, specificity 0.70). Interestingly, imaging information derived from [^18^F]FET PET data contributed most importantly to the classifier [97]. The first study about the diagnostic value of MET PET radiomics using a random forest classifier for differentiating between RN and recurrent brain tumor was conducted by Hotta et al. in a total of 44 brain lesions (gliomas and metastatic brain tumor). The diagnostic performance was also compared with that of conventional TBR value. The authors reported a sensitivity, specificity, and accuracy of radiomics with random forest classifier of 90.1%, 93.9%, and 92.2% respectively, significantly lower when compared to those of TBR evaluation with a cutoff value of 2.83 (sensitivity, 60.6%; specificity, 72.7%; accuracy 63.6%) [98]. More recently, integrated radiomics-based models were also evaluated using textural features extracted from postoperative [^18^F]FDG PET, [^11^C]MET PET, and MRI images by Wang et al. A total of 160 glioma patients were enrolled in the study as the whole cohort and were further distributed randomly to either the primary cohort or validation cohort to explore and verify the discrimination performance of the model between tumor recurrence and RN. Combined with clinical characteristics, an integrated diagnosis model by logistic regression was developed. Finally, the age, TBR_mean_ of [^18^F]FDG PET, TBR_max_ of [^11^C]MET PET, and other 12 textual features were shown to be significant contributors for discriminating tumor recurrence from RN (*p* < 0.001) both in primary and validation cohorts [99]. In Table 4, we describe the main characteristics of the studies regarding AI applications in glioma recurrence/differential diagnosis.

## 8. Conclusions

Over the last few years, advanced imaging tools have been developed to face the difficult clinical issue of differential diagnosis in glioma recurrence, representing a challenging scenario due to treatment-related changes, PSP, pseudoresponse, and the limitation of conventional imaging. Amino acid PET tracers were demonstrated to be reliable diagnostic tools thanks to their high tumor tissue uptake and low background in normal grey matter, giving additional and early information to standard modalities. Among them, MET PET seems to present the highest diagnostic value, but its use is limited to on-site cyclotron facilities. For this reason, [^18^F]labelled amino acid PET tracers, such as FDOPA, were developed, demonstrating comparable accuracy. Similarly, FET PET was revealed as suitable for clinical application thanks to its efficient radiosynthesis, also allowing unique kinetic analyses (dynamic acquisitions) enabling the assessment of TACs related to tumor grading/behaviour. Moreover, semiquantitative analyses provide useful information, increasing the accuracy of PET examinations. Notably, TBRs have been shown to be the most reliable PET parameters; however, a standardization in image acquisition techniques, machines, processing techniques, and analysis are warranted to reduce differences between studies. In this scenario, the growing knowledge about hybrid imaging PET/MRI and the application of artificial intelligence (AI) and machine learning (ML) could represent in the near future the turned key in the evaluation of glioma recurrence.

## Figures and Tables

**Figure 1 diagnostics-12-00844-f001:**
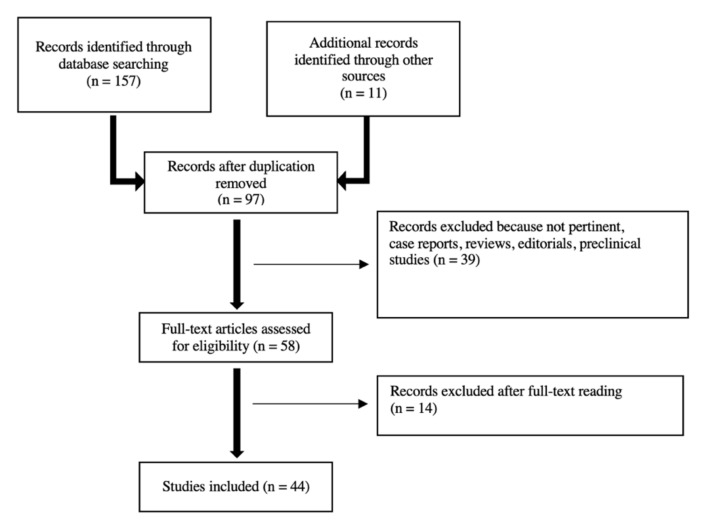
Studies’ workflow.

**Table 1 diagnostics-12-00844-t001:** Summary of the described studies regarding [^18^F]FET PET in glioma recurrence/differential diagnosis.

Authors [Ref.]	Year	Number of Patients	Glioma Grade (*n*)	PET Parameter	MRI/Other Imaging Modality Parameter	Main Findings
Galldiks et al. [35]	2015	124	55 grade II 19 grade III 50 grade IV	TBR_max_ TBR_mean_ TTP	CeMRI	Compared with the diagnostic accuracy of conventional MRI (85%) to diagnose tumor progression or recurrence, a higher accuracy (93%) was achieved by [^18^F]FET PET when a TBR_mean_ ≥ 2.0 or TTP < 45 min was present (sensitivity, 93%; specificity, 100%; accuracy, 93%; positive predictive value, 100%; *p* < 0.001).
Pyka et al. [36]	2018	47	3 grade II 16 grade III 27 grade IV	TBR TTP	rCBV ADC	Sensitivities and specificities for static PET were 80 and 85%, 66% and 77% for PWI, 62 and 77% for DWI, and 64 and 79% for PET TTP, respectively. Multiparametric analysis resulted in an AUC of 0.89, notably yielding a sensitivity of 76% vs. 56% for PET alone at 100% specificity.
Popperl et al. [37]	2006	45	26 grade II 7 grade III 12 grade IV	SUV_max_ TBR_max_ TTP	ND	TAC slightly and steadily increased in tumor-free patients and in LGG, whereas HGG showed an early peak around 10–15 min after injection followed by a decrease.
Maurer et al. [38]	2020	127	21 grade II 36 grade III 68 grade IV 2 ND	TBR_max_ TBR_mean_ TTP slope	ND	The highest accuracy for differentiating progression from TRCs was achieved by a combination of TBR_max_ and slope (sensitivity, 86%; specificity, 67%; accuracy, 81%). The accuracy of [^18^F]FET PET was higher in IDH-wildtype gliomas than in IDH-mutant ones (*p* < 0.001)
Bashir et al. [39]	2019	146	146 grade IV	TBR_max_ TBR_mean_ BTV	ND	TBR_max_ is a powerful imaging biomarker to detect recurrent GBM (sensitivity 99%, specificity 94%; *p* < 0.0001). BTV is independently and inversely correlated with OS.
Pöpperl et al. [40]	2004	53	27 grade IV 16 grade III 9 grade II 1 grade I	SUV_max_ TBR_max_	ND	Best differentiation between benign posttherapeutic effects and tumor recurrence was observed at a threshold value of 2.0 for the TBR, with a discriminatory power of 100%. For the absolute values of SUV_max_, the best differentiation was seen at a threshold value of 2.2.
Kebir et al. [41]	2016	26	26 grade IV	TBR_max_ TBR_mean_ TTP	ND	TBR_max_ and TBR_mean_were significantly higher in patients with true progression than in patients with late PSP, whereas TTP was significantly shorter. ROC analysis yielded an optimal cutoff value of 1.9 for TBR_max_ to differentiate between true progression and late PSP (sensitivity 84%, specificity 86%, accuracy 85%, *p* < 0.015).
Galldiks et al. [42]	2012	10	1 grade III 9 grade IV	TBR_max_ TBR_mean_ TTP	ND	A reduction in TBR_mean_ of ≥17% at follow-up differentiated responders (PFS ≥ 6 months) from non-responders (PFS < 6 months) with excellent sensitivity (83%) and specificity (100%). Moreover, TTP and kinetic patterns at baseline and follow-up differentiated responders from non-responders with a favourable diagnostic performance.
George et al. [43]	2018	13	13 grade IV	Dynamic acquisition	CeMRI	An only moderate correlation between FET PET uptake and CeMRI. FET PET may have a prognostic role in the follow-up of patients with recurrent GBM undergoing antiangiogenic therapy.
Hutterer et al. [44]	2011	11	11 grade IV	SUV_max_ TBR_max_	ND	In HGG patients undergoing antiangiogenic treatment, [^18^F]FET PET seems to be predictive for treatment failure.
Kertels et al. [45]	2019	36	36 grade IV	TBR *	ND	[^18^F]FET PET is a reliable tool for the detection of late PSP in GBM, irrespective of the analytical approach.
Steidl et al. [48]	2020	104	9 grade II 24 grade III 70 grade IV 1 other	TBR_max_ slope	rCBV_max_	The sensitivity of the rCBV_max_ was low (0.53), while the sensitivity of the combined TBR_max_and slope values was substantially higher (0.96). In the subgroup of IDH-mutant tumors, PWI appeared to be more reliable than [^18^F]FET PET.
Verger et al. [49]	2018	31	2 grade II 3 grade III 27 grade IV	TBR_max_ TBR_mean_ TTP slope	rCBF rCBV	TBR_max_was the only parameter that showed a significant diagnostic power to discriminate between TRC and progressive/recurrent gliomas. The best cutoff value for TBR_max_was 2.61, with a sensitivity of 80%, a specificity of 86%, a PPV of 95%, an NPV of 55%, and an accuracy of 81%. [^18^F]FET PET is superior to PWI for diagnosing progressive or recurrent gliomas.
GoÖttler et al. [50]	2016	30	3 grade II 4 grade III 23 grade IV	TBR_mean_ TTP slope	rCBV	Static and dynamic FET uptake measures and rCBV are interdependent and exhibit only a poor spatial overlap: the mean distance between the tumor hotspots of FET uptake and rCBV was 20.0 +/− 14.1 mm.
Lohmeier et al. [51]	2019	42	40 HGG 2 LGG	SUV_max_ SUV_mean_ TBR_max_ TBR_mean_	rADC_mean_	The ADC_mean_ in the metabolically most active regions was higher in patients with recurrent glioma than in patients with TRC. The highest accuracy (90%) was achieved when both DWI and [^18^F]FET PET-derived parameters were combined in a biparametric approach.
Sogani et al. [52]	2017	32	N.S.	TBR_max_ TBR_mean_	N rCBV ADC_mean_ Cho/Cr	The diagnostic accuracy, sensitivity, and specificity for recurrence detection using all three MRI parameters were 93.75%, 96%, and 85.7%, respectively. The addition of FET PET TBR values improved these values further to 96.87%, 100%, and 85.7%, respectively.
Jena et al. [53]	2016	26	N.S.	TBR_max_ TBR_mean_	N rCBV ADC_mean_ Cho/Cr	The diagnostic accuracy of [^18^F]FET PET/MRI TBR values for the correct identification of recurrence of brain gliomas reached 93.8% using TBR_max_ of 2.11 or greater and 87.5% using TBR_mean_ of 1.437 or greater. The highest accuracy (96.9%) was obtained when both the TBR_max_ was greater than 2.11 (or TBR_mean_ > 1.44) and the Cho/Cr ratio > 1.42.

Legend: PET, positron emission tomography; MRI, magnetic resonance imaging; SUV, standardized uptake value; TBR, tumor-to-background ratio; TTP, time to peak; TAC, time–activity curves; Ce, contrast enhancement; LGG, low-grade glioma; HHG, high-grade glioma; FET, fluoroethyl-tyrosine; PSP, pseudoprogression; N rCBV, normalized relative mean cerebral blood volume; Cho/Cr, choline-to-creatine; rADC, relative apparent diffusion coefficient; CBF, relative cerebral blood flow; TRC, treatment-related changes; PPV, positive predictive value; NPV, negative predictive value; DWI, diffusion weighted MRI; PWI, perfusion-weighted MRI; GBM, glioblastoma multiforme; BTV, biological tumor volume; OS, overall survival; ND, not determined or inconclusive. * different analytical approach (e.g., reference regions) were explored.

**Table 2 diagnostics-12-00844-t002:** Summary of the described studies regarding [^18^F]FDOPA PET in glioma recurrence/differential diagnosis.

Authors [Ref]	Year	Number of Patients	Glioma Grade (*n*)	PET Parameter	MRI/Other Imaging Modality Parameter	Main Findings
Herrmann et al. [56]	2014	110	33 grade III 77 grade IV	Visual analysis SUV_max_ SUV_mean_ TNR_max_ TSR_max_	ND	FDOPA PET showed a diagnostic accuracy of 82% (sensitivity, 89.6%; specificity, 72.4%) in distinguishing recurrence from TRC. Moreover, FDOPA PET is highly prognostic of PFS.
Zaragoni et al. [58]	2020	51	18 grade II 8 grade III 25 grade IV	TNR_max_ TSR_max_ MTV TTP	ND	All studied PET parameters, except TTP, were significant univariate predictors of glioma recurrence/progression (*p* < 0.001), with a global diagnostic accuracy of 96% being reached with TNR_max_, TSR_max_, and MTV. All PET parameters, except TTP, were also significant predictors of PFS, although none were predictive of OS
Karunanithi et al. [59]	2013	28	2 grade I 8 grade II 5 grade III 13 grade IV	SUV_max_ TNR_max_ TSR_max_ TWR_max_ TCR_max_	ND	The sensitivity, specificity and accuracy of [^18^F]FDG PET were 47.6%, 100%, and 60.7%, respectively, and those of [^18^F]FDOPA PET/CT were 100%, 85.7%, and 96.4%, respectively. The difference in the findings between [^18^F]FDG PET/CT and [^18^F]FDOPA PET/CT was significant (*p* = 0.0005). The difference was significant for LGGs but not for HGGs.
Karunanithi et al. [60]	2013	35	2 grade I 9 grade II 8 grade III 16 grade IV	SUV_max_ TNR_max_ TSR_max_ TWR_max_ TCR_max_	CeMRI	Comparison between CeMRI and [^18^F]FDOPA PET for detecting recurrent glioma showed a diagnostic accuracy of 80% vs. 97.1%, overall sensitivity 92.3% vs. 100%, and specificity 44.4% vs. 88.8%, respectively.
Youland et al. [61]	2018	13	2 grade II 4 grade III 7 grade IV	SUV_max_ SUV_mean_ TNR_max_	CeMRI	Regions of high PET avidity with an SUV_max_ > 1.36 or TNR_max_ > 2.0 had better sensitivity and specificity for tumor than CeMRI.
Cicone et al. [62]	2015	44	3 unverified 11 grade II 17 grade III 19 grade IV	Visual analysis TBR_mean_	rCBV	The regions with increased FDOPA uptake were much larger than those with increased rCBV values. In addition, TBR_mean_ is significantly higher for FDOPA uptake than for rCBV maps, indicating that PET is superior to PWI for differentiating between tumor and normal brain tissue.
Ledezma et al. [63]	2009	91	33 grade II 24 grade III 34 grade IV	Visual analysis	CeMRI	FDOPA detected most gliomas with sensitivity 95.2% (vs. MRI 90.5%), irrespective of tumor grade, labelling both enhancing and non-enhancing tumors equally well. FDOPA may be better at differentiating a non-enhancing tumor from other causes of MRI-T2w signal change such as gliosis and oedema.
Bund et al. [64]	2017	53	35 LGG 18 HGG	SUV_max_ TNR_max_	Cho/Cr Cho/NAA	Significant correlation between FDOPA SUV_max_and the MRS ratios was shown, which correspond to the proliferative and infiltrative characteristics of the tumor, respectively. A threshold of 2.16 in TNR at 30 min is useful to discriminate LGGs and HGGs.
Karavaeva et al. [65]	2015	29	9 grade III 20 grade IV	SUV_mean_	ADC	Areas of high [^18^F]FDOPA uptake exhibited low ADC, and areas of hyperintensity T2/FLAIR with low [^18^F]FDOPA uptake exhibited high ADC. Median [^18^F]FDOPA uptake was positively correlated, and median ADC was inversely correlated with mitotic index from resected tumor tissue.

Legend: Ce, contrast enhancement; MRI, magnetic resonance imaging; FDOPA, fluoro-l-dihydroxy-phenylalanine; PET, positron emission tomography; SUV, standardized uptake value; TNR, tumor to contralateral normal hemispheric brain tissue ratio; TSR, tumor to normal striatum ratio; TWR, tumor to normal white matter ratio; TCR, tumor to normal cerebellum ratio; FDG, Fluorodeoxyglucose; LGG, low-grade glioma; HHG, high-grade glioma; TRC, treatment-related changes; rADC, relative apparent diffusion coefficient; PFS, progression-free survival; FLAIR, fluid-attenuated inversion recovery; rCBV, relative mean cerebral blood volume; MRS, magnetic resonance spectroscopy; Cho/Cr, choline-to-creatine; Cho/NAA, choline-to-N-Acetyl-Aspartate; MTV, metabolic tumor volume; TTP, time to peak; OS, overall survival; ND, not determined or inconclusive.

**Table 3 diagnostics-12-00844-t003:** Summary of the described studies regarding [^11^C]MET PET in glioma recurrence/differential diagnosis.

Authors [Ref]	Year	Number of Patients	Glioma Grade (*n*)	PET Parameter	MRI/Other Imaging Modality Parameter	Main Findings
Grosu et al. [28]	2011	29/42	1 grade I 2 grade II 11 grade III 14 grade IV	SUV_mean_ TBR_mean_	ND	FET PET and MET PET provide comparable diagnostic information with a sensitivity of 91% and specificity of 100% for both radiotracers.
Jung et al. [66]	2016	42	12 grade III 30 grade IV	TBR_max_ TBR_mean_ MTV	ND	TBR and MTV had a diagnostic value to differentiate recurrence from posttreatment effect. Unlike TBR, MTV was shown to be an independent factor in patients with recurrence.
Tsuyuguchi et al. [69]	2004	11	3 grade III 8 grade IV	Visual analysis, SUV_mean_ TBR_mean_	ND	MET PET reached a sensitivity, specificity, and accuracy in detecting tumor recurrence of 100%, 60%, and 82%, respectively.
D’Souza et al. [70]	2014	29	16 grade III 12 grade IV	SUV_max_ SUV_mean_	rCBV Cho/Cr Cho/NAA	The sensitivity, specificity, and accuracy of MET PET in identifying tumor recurrence/residual were 94.7%, 80%, and 89.6%, respectively, whereas those of MRI were 84.2%, 90%, and 86.2%, respectively.
Minamimoto et al. [71]	2015	31/70	12 grade III 19 grade IV	Visual analysis, SUV_max_ SUV_mean_ TBR_max_ TBR_mean_	ND	The TBR_max_ and TBR_mean_ was significantly higher for tumor recurrence than for radiation-induced necrosis (*p* < 0.02). The visual assessment showed no significant difference from the quantitative assessment of MET PET with a relevant cutoff value for the differentiation of recurrent brain tumors from radiation-induced necrosis.
Kits et al. [72]	2018	23/30	5 grade II 8 grade III 10 grade IV	TBR_mean_cortex TBR_mean_mirror TBR_max_cortex TBR_max_mirror	ND	Clinically relevant cutoffs were TBR_max_mirror ≥ 1.99 giving a specificity of 100% for tumor recurrence with a sensitivity of 76% and TBR_max_cortex ≥ 1.58 giving a sensitivity and specificity of 90 and 78%, respectively.
Deuschl et al. [73]	2017	50	14 grade II 16 grade III 20 grade IV	SUV_max_ SUV_mean_ TBR_max_ TBR_mean_	CeMRI	Diagnostic accuracy was 82% for MRI, 88% for [^11^C]MET PET, and 96% for hybrid [^11^C]MET PET/MRI.
Terakawa et al. [74]	2008	26/77	6 grade II 6 grade III 14 grade IV	SUV_max_ SUV_mean_ TBR_max_ TBR_mean_	ND	TBR_mean_ value seems to provide the best sensitivity and specificity in differentiating glioma recurrence from RN.
Shishido et al. [75]	2012	21	8 grade III 13 grade IV	SUV_max_ SUV_mean_ TBR_max_ TBR_mean_	ND	The average TBR of recurrent gliomas was significantly higher than that of necrotic lesions on MET PET (*p* < 0.01).
Tripathi et al. [76]	2012	37	2 grade I 13 grade II 8 grade III 12 grade IV	SUV_max_ TBR_max_	ND	Using a cutoff for TBR_max_ > 1.9 to differentiate recurrence from no recurrence, the sensitivity of MET was 94.7%, whereas specificity was 88.89%.
Dandois et al. [77]	2010	28	14 grade III 14 grade IV	ND	rCBV	rCBV reached equal performances in differentiating tumor recurrence and RN than MET PET. Cutoff value of rCBV for differentiating tumor from necrosis was 182% (sensitivity, 81.5%; specificity, 100%).
Qiao et al. [78]	2019	33	10 grade III 23 grade IV	SUV_max_ SUV_mean_ TBR_max_ TBR_mean_	rCBV_mean_	Combining the assessment of TBR_max_ and TBR_mean_ and relative rCBV_mean_, the highest sensitivity (0.848) and specificity (1.0) was shown.

Legend: MET, methyl-l-methionine; SUV, standardized uptake value; TBR, tumor-to-background ratio; RN, radiation necrosis; rCBV, relative mean cerebral blood volume; FET, fluoroethyl-tyrosine; Cho/Cr, choline-to-creatine; Cho/NAA, choline-to-N-acetyl-aspartate; MTV, metabolic tumor volume; PET, positron emission tomography; Ce, contrast enhancement; MRI, magnetic resonance imaging; ND, not determined or inconclusive.

**Table 4 diagnostics-12-00844-t004:** Summary of the described studies regarding PET artificial intelligence application in glioma recurrence/differential diagnosis.

Authors [Ref.]	Patients	WHO Grade	RF	Classification Model	Accuracy
Kebir et al. [94]	14	III/IV	[^18^F]FET	Unsupervised consensus clustering	75%
Kebir et al. [95]	44	IV	[^18^F]FET	linear discriminant analysis	AUC 93%
Lohmann et al. [96]	34	IV	[^18^F]FET	random forest	70%
Paprottka et al. [97]	66	I-IV	[^18^F]FET	random forest	86%
Hotta et al. [98]	41	ND	[^11^C]MET	random forest	92.2%
Wang et al. [99]	160	II/III/IV	[^11^C]MET	random forest	AUC 93.2%

Legend: RF, radiopharmaceutical; FET, fluoroethyl-tyrosine; MET, methyl-l-methionine; AUC, area under the curve; ND, not determined or inconclusive.

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
