# Peer review of "The Utility of Conventional Amino Acid PET Radiotracers in the Evaluation of Glioma Recurrence also in Comparison with MRI"

_diagnostics, 2022, doi:10.3390/diagnostics12040844_

Round 1
Reviewer 1 Report
Authors summarized all the important aminoacid-based radiotracers in detail --both routinely used ones and novel ones.
However, a significant portion of new research is missing on amino acid transport systems, including System L, System A, and System ASC. Cationic, anionic amino acid transporters are heavily published (work from McConathy, Goodman, et al). For example, [18F]AFETP radiotracer role with respect to arginine in several carcinoma models.
Author Response
We sincerely thank both reviewers for their time and suggestions. The improved manuscript will be in track change, as requested by MDPI.
Reviewer 1
Authors summarized all the important aminoacid-based radiotracers in detail --both routinely used ones and novel ones.
However, a significant portion of new research is missing on amino acid transport systems, including System L, System A, and System ASC. Cationic, anionic amino acid transporters are heavily published (work from McConathy, Goodman, et al). For example, [18F]AFETP radiotracer role with respect to arginine in several carcinoma models.
A1) Thank you for this comment. We added a dedicated paragraph accordingly.
Reviewer 2 Report
This is an excellent and timely "review' on a subject that has been under the radar for a long time, The usefulness of amino acid based radiopharmaceuticals and PET imaging for delineating brain tumours to some of us has been apparent for a while, and used even on a routine basis- or should I say when the radiotracers are available. Hence, it is very helpful to see positive reviews such as this to assist with the adoption of these tracers (particularly with18F-FET) more widely.
However, I believe availability and cost may be a deterrent in some areas. Hence published trials, reviews etc showing the value of these tracers would assist in their funding, accessibility and use.
The review is well written, researched and described.
The manuscript only requires minor English and grammatical checks.
Author Response
We sincerely thank both reviewers for their time and suggestions. The improved manuscript will be in track change, as requested by MDPI.
Reviewer 2
This is an excellent and timely "review' on a subject that has been under the radar for a long time, The usefulness of amino acid based radiopharmaceuticals and PET imaging for delineating brain tumours to some of us has been apparent for a while, and used even on a routine basis- or should I say when the radiotracers are available. Hence, it is very helpful to see positive reviews such as this to assist with the adoption of these tracers (particularly with18F-FET) more widely.
However, I believe availability and cost may be a deterrent in some areas. Hence published trials, reviews etc showing the value of these tracers would assist in their funding, accessibility and use.
The review is well written, researched and described.
The manuscript only requires minor English and grammatical checks.
A2) Thank you for these constructive, supportive words and comments. We now extensively improved the whole manuscript English.